# Risk of Seizure Aggravation after COVID-19 Vaccinations in Patients with Epilepsy

**DOI:** 10.3390/vaccines12060593

**Published:** 2024-05-30

**Authors:** William C.Y. Leung, Ryan Wui-Hang Ho, Anthony Ka-Long Leung, Florinda Hui-Ning Chu, Cheuk Nam Rachel Lo, Andrian A. Chan, Cheuk Yan Claudia Chan, Desmond Yin Hei Chan, Jacklyn Hoi Ying Chui, Wai Tak Victor Li, Elton Hau Lam Yeung, Kay Cheong Teo, Gary Kui-Kai Lau, Richard Shek-Kwan Chang

**Affiliations:** 1Division of Neurology, Department of Medicine, Queen Mary Hospital, University of Hong Kong, Hong Kong SAR, China; ryanhowh@connect.hku.hk (R.W.-H.H.); kalongleung@gmail.com (A.K.-L.L.); florindachu@gmail.com (F.H.-N.C.); rachello818@gmail.com (C.N.R.L.); kcteo@hku.hk (K.C.T.); gkklau@hku.hk (G.K.-K.L.); 2School of Clinical Medicine, Li Ka Shing Faculty of Medicine, University of Hong Kong, Hong Kong SAR, China; andrianc@connect.hku.hk (A.A.C.); ccheukyanchan@gmail.com (C.Y.C.C.); descyh@connect.hku.hk (D.Y.H.C.); jackchui@connect.hku.hk (J.H.Y.C.); victorliwaitak.clinical@gmail.com (W.T.V.L.); yeungelt@connect.hku.hk (E.H.L.Y.)

**Keywords:** epilepsy, seizure, COVID-19 vaccine, COVID-19 infection, vaccination gap, neurological adverse effects

## Abstract

Although Coronavirus disease 2019 (COVID-19) vaccinations are generally recommended for persons with epilepsy (PwE), a significant vaccination gap remains due to patient concerns over the risk of post-vaccination seizure aggravation (PVSA). In this single-centre, retrospective cohort study, we aimed to determine the early (7-day) and delayed (30-day) risk of PVSA, and to identify clinical predictors of PVSA among PwE. Adult epilepsy patients aged ≥18 years without a history of COVID-19 infection were recruited from a specialty epilepsy clinic in early 2022. Demographic, epilepsy characteristics, and vaccination data were extracted from a centralized electronic patient record. Seizure frequency before and after vaccination, vaccination-related adverse effects, and reasons for or against vaccination were obtained by a structured questionnaire. A total of 786 PwEs were included, of which 27.0% were drug-resistant. At the time of recruitment, 74.6% had at least 1 dose of the COVID-19 vaccine. Subjects with higher seizure frequency (*p* < 0.0005), on more anti-seizure medications (*p* = 0.004), or had drug-resistant epilepsy (*p* = 0.001) were less likely to be vaccinated. No significant increase in seizure frequency was observed in the early (7 days) and delayed phases (30 days) after vaccination in our cohort. On the contrary, there was an overall significant reduction in seizure frequency 30 days after vaccination (1.31 vs. 1.89, *t* = 3.436; *p* = 0.001). This difference was seen in both types of vaccine (BNT162b2 and CoronaVac) and drug-resistant epilepsy, but just missed significance for the second dose (1.13 vs. 1.87, *t* = 1.921; *p* = 0.055). Only 5.3% had PVSA after either dose of vaccine. Higher pre-vaccination seizure frequency of ≥1 per week (OR 3.01, 95% CI 1.05–8.62; *p* = 0.04) and drug-resistant status (OR 3.32, 95% CI 1.45–249 7.61; *p* = 0.005) were predictive of PVSA. Meanwhile, seizure freedom for 3 months before vaccination was independently associated with a lower risk of PVSA (OR 0.11, 95% CI 0.04–0.28; *p* < 0.0005). This may guide epilepsy treatment strategies to achieve better seizure control for at least 3 months prior to vaccination. As COVID-19 shifts to an endemic phase, this study provides important data demonstrating the overall safety of COVID-19 vaccinations among PwE. Identification of high-risk patients with subsequent individualized approaches in treatment and monitoring strategies may alleviate vaccination hesitancy among PwE.

## 1. Introduction

Despite the World Health Organization declaring the end of Coronavirus disease 2019 (COVID-19) as a public health emergency in May 2023, it remains a significant burden on the global healthcare systems as an endemic disease [1,2]. Vaccination against COVID-19 is highly effective in reducing hospitalizations [3], risk of severe COVID-19, and mortality [4].

Persons with epilepsy (PwE) are at an increased risk of COVID-19 infection [5]. COVID-19 infection may also worsen seizure control among PwE, which may be secondary to direct viral CNS invasion, systemic cytokine release, or hypoxia-related encephalopathy [6]. Febrile seizures may also occur in COVID-19 in certain epilepsy syndromes [7]. Although the International League Against Epilepsy (ILAE) has recommended epilepsy patients to be vaccinated against COVID-19 [8], there is a low vaccine uptake rate among PwE ranging from 17 to 70% in previous studies [9,10,11,12]. This vaccination gap may largely be due to a subjective fear of seizure aggravation, which remains an important concern and a significant barrier to vaccination among this population [13].

In early 2021, two COVID-19 vaccines were introduced to the public in Hong Kong, namely the inactivated virus vaccine (CoronaVac) and the messenger RNA (mRNA) vaccine (BNT162b2). A recent territory-wide study in Hong Kong has reported a low risk of post-vaccination seizures in the general population [14]. However, there are limited data worldwide on the risk of seizure aggravation among PwE.

We conducted a single-centre retrospective observational study to determine the early (7-day) and delayed (30-day) risk of seizure exacerbation after COVID-19 vaccination among PwE in Hong Kong. We also aimed to evaluate the COVID-19 vaccination uptake rate and the cues for and against vaccination in PwE. Finally, we aimed to identify clinical predictors in epilepsy that are associated with the risk of post-vaccination seizure aggravation. 

## 2. Materials and Methods

### 2.1. Study Design

This was a single-centre, retrospective, observational cohort study in Hong Kong SAR, China. All patients aged ≥18 who attended a follow-up appointment at an epilepsy specialty outpatient clinic between 1 January 2021 and 31 December 2021 were recruited. Subject recruitment was conducted from January to March 2022. Due to social restrictions during the COVID-19 pandemic, recruitment was performed either in person during follow-up clinic visits or over phone interviews. To ensure adequate representation of patients with severe epilepsy who may not be fit for verbal consent, the main caregiver was interviewed after consent had been obtained from the subject’s next of kin. Subjects with a history of COVID-19 infection before 1 January 2021 were excluded.

Baseline characteristics were extracted from a centralized electronic patient record of the Hospital Authority of Hong Kong. Patient demographics, past medical history, and medication records were obtained. Epilepsy characteristics including age at diagnosis, seizure subtypes, epilepsy etiology, number and dosage of anti-seizure medications (ASMs), drug-resistant status, and history of epilepsy surgery were recorded. The diagnosis of epilepsy was reviewed and confirmed by a neurologist based on clinical history according to the 2014 International League Against Epilepsy (ILAE) definition [15]. Relevant investigation results including neuroimaging and electroencephalography (EEG) were reviewed. 

Seizure information was obtained by a structured questionnaire, which was completed in person or over phone interviews by the research team. Missing data were supplemented by historical clinical notes from hospital admissions and outpatient clinics. The number of minor seizures, major seizures, and seizures requiring hospitalization 7 days before, 7 days (early) and 30 days (delayed) after each vaccination were recorded. To ensure at least 3 months of baseline seizure data prior to vaccination were included, the monthly frequency of all types of seizures was recorded for all subjects from November 2020 until the date of recruitment.

COVID-19 vaccination data were obtained from a centralized territory-wide vaccination database in the electronic patient record of Hospital Authority Hong Kong. This was supplemented by patient-reported vaccinations administered outside of Hong Kong. The date and type of COVID-19 vaccination and any history of COVID-19 infection were obtained. Self-reported vaccination-related adverse effects related and unrelated to epilepsy were recorded. Reasons for or against COVID-19 vaccination were collected from all subjects by a structured questionnaire. Details of the questionnaire are reported in Appendix A.

### 2.2. Definitions

Epilepsy was defined as (1) at least two unprovoked seizures occurring >24 h apart, (2) one unprovoked seizure and a probability of further seizures similar to the general recurrence risk (at least 60%) after two unprovoked seizures occurring over the next 10 years, or (3) diagnosis of an epilepsy syndrome [15]. Drug-resistant epilepsy was diagnosed as per the ILAE definition as failure of adequate trials of two tolerated and appropriately chosen and used anti-seizure medication schedules to achieve sustained seizure freedom [16]. Seizure semiology was classified according to the 2017 ILAE Operational Classification as “focal onset”, “generalized onset”, or “unclassified or unknown” [17]. “Minor seizure attacks” were defined as self-limiting focal aware seizures or focal impaired awareness seizures, while “major seizure attacks” were defined as generalized tonic-clonic seizures, or intractable seizures requiring rescue anti-seizure therapy or hospitalizations. Epilepsy surgery was defined as epilepsy resective surgeries and neuromodulation device therapies (including vagus nerve stimulation and deep brain stimulation).

“Usual seizure frequency” was calculated among all recruited subjects and refers to the average monthly seizure frequency from January to December 2021. “Pre-vaccination seizure frequency (PVSF)” was calculated among all vaccinated subjects and refers to the average monthly seizure frequency in 3 months prior to the first dose of COVID-19 vaccination. “Post-vaccination seizure aggravation (PVSA)” was defined as any increase in post-vaccination seizure frequency compared with the corresponding period prior to vaccination.

### 2.3. Statistical Analysis

Statistical analyses were performed using SPSS version 26.0 (SPSS Inc., USA). Continuous variables were expressed as mean and standard deviation, while categorical variables were presented by numbers and percentages (%). Statistical significance was examined with 2-tailed tests, with statistical significance defined at *p* < 0.05.

Baseline characteristics including patient demographics and epilepsy data were compared between vaccinated and unvaccinated subjects to delineate possible predictors for vaccine uptake using Pearson’s *χ*^2^ (for categorical variables) and independent sample *t* tests (for continuous variables). Early change in post-vaccination seizure frequency was determined by comparing the number of seizures 7 days before and after each vaccination dose using a paired sample *t* test. Delayed change in post-vaccination seizure frequency was determined by comparing the number of seizures 30 days after each vaccination dose and PVSF using a paired sample *t* test. 

To further determine possible risk factors of PVSA, univariate logistic regression analysis was performed. Age, sex, and epilepsy characteristics including time from diagnosis, seizure classification, epilepsy etiology, medication refractoriness, history of epilepsy surgery, number of anti-seizure medications, and PVSF were separately entered into univariate logistic regression models as predictors. We then included all variables with a univariable *p* < 0.1 in the multivariate analysis and used a backward stepwise elimination approach to delineate their individual effects on PVSA.

## 3. Results

### 3.1. Study Population

A total of 1537 adult epilepsy patients were considered for enrollment, among which 11 subjects were excluded according to the exclusion criteria. We further excluded 260 patients who did not consent to the study and 480 subjects who did not complete the whole interview. A resulting cohort of 786 subjects consented to participate in the study, with a response rate of 51.5%.

The baseline characteristics of the final cohort are summarized in Table 1. The mean age at recruitment was 51.3 years, with a slight male predominance (54.2%). The majority of the cohort was of Chinese ethnicity (96.0%). The mean time from diagnosis of epilepsy to the date of recruitment was 18.9 years. Most subjects had focal onset seizures (58.8%), followed by generalized onset (27.4%) and unknown/unclassified onset seizures (17.2%). Structural etiology was identified in 34.4% of subjects. Other epilepsy etiologies included genetic (8.9%), infectious (2.4%), immune (1.1%), and metabolic (0.6%) causes.

A majority (79.2%) of the cohort had a usual seizure frequency of less than once per month, while 57.5% were seizure-free. On the other hand, 10.6% of subjects had more than one seizure per month and 10.2% of subjects had more than one seizure per week. Most subjects (59.2%) were on one ASM, while 21.5% and 10.6% were on two and more than two ASMs, respectively. Drug-resistant epilepsy accounted for 27.0% of the study cohort, while 36 subjects (4.7%) had a history of epilepsy surgery.

### 3.2. Patterns of Vaccination among People with Epilepsy

At the time of recruitment, 586 (74.6%) had received at least one dose of COVID-19 vaccination, among which 131 (16.7%), 354 (45.0%) and 101 (12.8%) received one, two and three doses (including booster dose), respectively. In the vaccinated group, 352 (60.3%) patients received BNT162b2 and 231 (39.6%) received CoronaVac. No subjects received more than one type of vaccine.

We compared the baseline characteristics to look for any patterns of vaccination within the cohort (Table 1). The vaccinated and unvaccinated groups did not differ significantly in terms of age, sex, and ethnicity. Epilepsy characteristics including time from epilepsy diagnosis, seizure onset classification, and epilepsy etiology were similar across both groups. Subjects were more likely to be vaccinated if they were seizure-free at baseline (*p* = 0.015). On the contrary, subjects with higher seizure frequency (*p* < 0.0005) or on more ASMs were less likely to be vaccinated (*p* = 0.005). Subjects with drug-resistant epilepsy were also less likely to be vaccinated (*p* = 0.001). 

### 3.3. Cues for and against Vaccination

We studied the reasons for and against vaccination in our cohort (Figure 1). Among the vaccinated group, protection from COVID-19 infection (74.2%) and protection of others from infection (46.9%) were the most common cues for vaccination. Community infective control measures to encourage compliance with government vaccination requirements were a common reason for vaccination (40.4%). A proportion of subjects decided for vaccination due to the perceived safety of the vaccine (29.4%), while 23.2%, 13.1%, and 5.3% had encouragement from healthcare providers, media or social media, and family members, respectively.

Among the 200 subjects who had not received any COVID-19 vaccination at the time of recruitment, most subjects decided against vaccination due to fear of post-vaccine seizure aggravation (59.0%) and fear of uncertain side effects (50.5%). Discouragement from healthcare workers (21.0%), perceived lack of scientific evidence (15.0%), perceived low severity of COVID-19 infection (9.5%), and perceived low prevalence of COVID-19 in Hong Kong (5.5%) were among the cues against vaccination.

### 3.4. Post-Vaccination Adverse Effects

Self-reported post-vaccination adverse effects were recorded for all vaccinated subjects (Table 2). Among 586 vaccinated subjects, 36.7% of subjects did not report any post-vaccination adverse effects. Most subjects reported mild general adverse effects after vaccination, including local injection site reactions (42.2%), malaise (18.1%), headache (8.4%), and myalgia (6.1%). Fever occurred in 11.1% of subjects, while 2.9% experienced chills. Meanwhile, 2.6% of subjects experienced a subjective increase in minor seizure attacks, while 0.7% reported a subjective increase in major attacks. 

### 3.5. Seizure Frequency before and after COVID-19 Vaccination

No significant change in seizure frequency was noted in the early phase (7 days) after the first dose (0.33 vs. 0.33, *t* = 0.107; *p* = 0.914) and the second dose of vaccination (0.26 vs. 0.29, *t* = 1.118; *p* = 0.264) compared to 7 days prior to each vaccination (Table 3). The results were similar in both types of vaccination (BNT162b2 and CoronaVac). Interestingly, subjects with drug-resistant epilepsy experienced a slight but statistically significant reduction in number of seizures after the first vaccination dose (0.90 vs. 1.08, *t* = 2.236; *p* = 0.027), but not after the second dose (0.87 vs. 0.98, *t* = 1.181; *p* = 0.241).

After excluding subjects with incomplete pre-vaccination seizure frequency (PVSF) data, we compared the number of seizures in the delayed phase (30 days) after vaccination with the PVSF (*n* = 489, Table 3). A significant reduction in seizure frequency was noted 30 days after the first vaccination dose in all subjects (1.31 vs. 1.89, *t* = 3.436; *p* = 0.001). There was also an overall trend toward reduced seizure frequency after the second vaccination dose approaching statistical significance (1.13 vs. 1.87, *t* = 1.921; *p* = 0.055). This observation was similar in both types of vaccination (BNT162b2 and CoronaVac), and in subjects with drug-resistant epilepsy (4.35 vs. 6.18, *t* = 3.099; *p* = 0.002). 

Four subjects in our cohort had seizures requiring hospitalization within 30 days after vaccination. All four occurred after the second vaccination dose (three after BNT162b2 and one after CoronaVac). Upon review of case records, poor compliance to anti-seizure medications and recent alcohol intake were reported in two subjects as possible contributing factors. None of the four subjects developed status epilepticus or required admission into the intensive care unit. All four subjects received the third booster COVID-19 vaccination dose uneventfully afterward without any documented post-vaccination seizures.

### 3.6. Predictors for Post-Vaccination Seizure Aggravation

Post-vaccination seizure aggravation (PVSA) was observed in 26 subjects (5.3%), of which 18 (3.7%) and 17 (4.4%) reported increased seizure frequency after the first and second vaccination doses, respectively. Univariate logistic regression analysis was performed to identify possible risk factors of PVSA (Table 4). Pre-vaccination seizure frequency of ≥1 per week predicted a higher risk of PVSA (OR 3.01, 95% CI 1.05–8.62; *p* = 0.04). Subjects with drug-resistant epilepsy were also more likely to have PVSA (OR 3.32, 95% CI 1.45–7.61; *p* = 0.005). On the other hand, seizure freedom in 3 months prior to vaccination was predictive of a lower risk of PVSA (OR 0.10, 95% CI 0.04–0.25; *p* < 0.0005). 

Further multivariate analysis showed that seizure freedom for 3 months was independently associated with a lower risk of PVSA (OR 0.11, 95% CI 0.04–0.28; *p* < 0.0005).

## 4. Discussion

Previous studies have reported an overall low risk of COVID-19 vaccination-related seizure aggravation among PwE [10,18,19,20,21,22]. A recent meta-analysis showed that only a small proportion of vaccinated patients with epilepsy had an increase in seizure frequency [23]. However, most studies were limited by relatively small sample sizes. Moreover, seizure aggravation was often self-reported as an umbrella phenomenon and lacked correlation with detailed quantitative seizure frequency data. This remains an important clinical question for epileptologists and may lead to vaccination hesitancy among epilepsy patients who are vulnerable to COVID-19 infections and complications [5]. To supplement currently available evidence, we conducted a large-scale observational study focusing on adult patients with epilepsy. 

### 4.1. Effect of COVID-19 Vaccinations on Seizure Frequency

No significant increase in seizure frequency was observed in the early phase (7 days) and delayed phase (30 days) after vaccination in our cohort. In fact, there was an overall significant reduction in seizure frequency 30 days after vaccination. This was similar in both types of COVID-19 vaccines (BNT162b2 and CoronaVac). Among subjects with drug-resistant epilepsy, reduced seizure frequency was observed 7 days and 30 days after the first dose of the vaccine. This is contrary to the usual understanding that vaccinations would generally lower seizure thresholds. This finding is consistent with a previous study that showed an 80% reduction in seizure frequency after COVID-19 vaccination [20]. However, this may be confounded by transient improvement in medication compliance due to fear of seizure aggravation, or lifestyle changes after vaccination (e.g., more rest and sleep). Future large-scale controlled studies may further validate this observation. Animal studies may also reveal any biological mechanisms behind this observation. 

Post-vaccination seizure aggravation (PVSA) was observed in 5.3% of subjects in our cohort, which is consistent with the findings in a previous meta-analysis [24]. Of note, one subject had a PVSA requiring hospitalization 30 days after the second dose of BNT162b2 vaccination, without status epilepticus or requirement of intensive care. Upon further chart review, this subject later received the third COVID-19 booster vaccination uneventfully without documented PVSA. 

Higher pre-vaccination seizure frequency of ≥1 per week in the 3 months preceding vaccination was associated with a higher risk of PVSA. Meanwhile, seizure freedom for 3 months prior to vaccination was associated with a lower risk of PVSA. This trend was consistent with previous studies [21,25]. This finding was also independent of drug-resistant status, epilepsy classification and etiology. This may guide healthcare providers to adjust treatment strategies to achieve better seizure control for at least 3 months prior to vaccination. 

Although subjects with drug-resistant epilepsy were 3 times more likely to have PVSA, there is an overall reduction in post-vaccination seizure frequency across the group as discussed above. Nonetheless, this suggests a need for increased vigilance of PVSA in this population.

### 4.2. Causes of Vaccination Hesitancy among Persons with Epilepsy 

Most unvaccinated subjects in our cohort were hesitant against vaccination due to fear of seizure aggravation (59.0%) and perceived lack of scientific evidence (15.0%). In fact, a significant proportion of subjects felt discouragement against vaccination from healthcare providers (21.0%). This could be due to a relative lack of evidence at the time of recruitment in early 2022 and could be addressed by the increasing clarity and availability of real-world data among PwE in recent years. Identification of patients at risk of PVSA and subsequent individualized treatment and monitoring strategies may address such patient concerns. 

Half of the subjects reported fear of uncertain general side effects of the vaccine (50.5%). As shown in our study, only mild general adverse effects were reported after vaccination. The incidence of post-vaccination general adverse effects was similar to previous studies in epilepsy and non-epilepsy populations [10,26].

A small but significant proportion of our subjects did not undergo vaccination as the caregiver was vaccinated instead (11.0%). This was especially apparent among subjects with severe epilepsy with dependent activities of daily living. Given the highly contagious nature of the COVID-19 virus, carer education on the risks and benefits of vaccination for PwE is of high importance.

### 4.3. Strengths and Limitations

To the best of the authors’ knowledge, this is the largest adult epilepsy cohort to date studying the risks of COVID-19 vaccination-related seizure aggravations. Drug-resistant epilepsy was well-represented in our cohort (27%), which is similar to the general prevalence among patients with epilepsy [27]. The cohort was well-characterized with detailed clinical data, epilepsy characteristics, and investigation results available from a centralized electronic patient database. Vaccination data including dates and type of vaccine were available for all subjects. To avoid selection bias against patients with severe epilepsy, caregivers were interviewed for subjects who were otherwise not mentally fit for consent.

There are some limitations to this study. First, the retrospective nature of self-reported seizure data means that they may be subject to recall bias. Subjects may not be able to recognize subtle seizures and non-motor manifestations of seizures. This was minimized by complementing these data by reviewing clinical and hospitalization notes for any unreported seizures. Second, subjects who were unable to recall complete PVSF data were excluded from the analysis. Third, some social demographic data, including educational background and marital status, were not available in our data and may be explored in future studies. Fourth, as most PVSA were self-reported and did not present to the hospital, the diagnosis of seizure could not be supplemented by objective investigations such as electroencephalography. This may have led to an overestimation of seizure occurrence after vaccination. This was minimized by reviewing the diagnosis through clinical history and outpatient records. Lastly, this study focused on any early (7 days) or delayed (30 days) change in seizure frequencies after vaccination. Further studies are warranted to determine any long-term change in seizure control among PwE. 

## 5. Conclusions

This study provided real-world observational data on the occurrence of seizure aggravation after COVID-19 vaccination. Our findings suggest that COVID-19 vaccinations do not cause an overall significant post-vaccination seizure aggravation among PwE. Epilepsy patients who have high seizure frequency within 3 months before vaccination or are drug-resistant are at higher risk of PVSA. Meanwhile, seizure freedom within 3 months is independently associated with a lower risk of PVSA. This may guide epilepsy management strategies in seizure control for at least 3 months prior to vaccination. Future large-scale, long-term, prospective controlled studies may further validate the observation of overall reduced seizure frequency after COVID-19 vaccination among PwE.

As COVID-19 shifts to an endemic phase, it is of essence to clarify the safety of COVID-19 vaccines in epilepsy. With the increasing availability of vaccination data in recent years, identification of high-risk patients with subsequent individualized approaches in treatment and monitoring strategies may alleviate vaccination hesitancy among PwE. 

## Figures and Tables

**Figure 1 vaccines-12-00593-f001:**
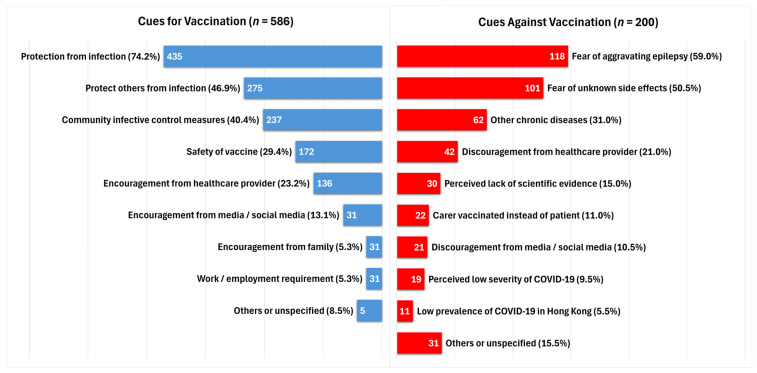
Cues for and against COVID-19 vaccination.

**Table 1 vaccines-12-00593-t001:** Comparison of demographics and epilepsy characteristics between vaccinated and unvaccinated subjects.

Variables	Total (*n* = 786) (Mean ± SD, *n* (%))	Vaccinated (*n* = 586) (Mean ± SD, *n* (%))	Unvaccinated (*n* = 200) (Mean ± SD, *n* (%))	*p* ^a^
Demographics				
Age, years	51.3 ± 18.3	51.0 ± 17.7	52.1 ± 19.9	0.480
Gender				0.224
Female	360 (45.8)	261 (44.5)	99 (49.5)	
Male	426 (54.2)	325 (55.5)	101 (50.5)	
Ethnicity				0.216
Chinese	719 (96.0)	526 (95.5)	193 (97.5)	
Non-Chinese	30 (4.0)	25 (4.5)	5 (2.5)	
Epilepsy characteristics				
Time from diagnosis, years	18.9 ± 14.0	19.0 ± 14.0	18.6 ± 14.0	0.745
Seizure onset				
Focal onset	462 (58.8)	351 (59.9)	111 (55.5)	0.275
Generalized onset	215 (27.4)	157 (26.8)	58 (29.0)	0.545
Unknown/unclassified	135 (17.2)	96 (16.4)	39 (19.5)	0.313
Epilepsy etiology				
Structural	270 (34.4)	198 (33.8)	72 (36.0)	0.570
Genetic	70 (8.9)	51 (8.7)	19 (9.5)	0.733
Others ^b^	33 (4.2)	21 (3.6)	12 (6.0)	0.141
Cryptogenic	400 (50.9)	308 (52.6)	92 (46.0)	0.109
Usual seizure frequency ^c^				<0.0005 *
<1/month	605 (79.2)	473 (82.4)	132 (69.5)	
≥1/month	81 (10.6)	48 (8.4)	33 (17.4)	
≥1/week	78 (10.2)	53 (9.2)	25 (13.2)	
Number of ASMs				0.005 *
0	69 (8.8)	55 (9.4)	14 (7.0)	
1	465 (59.2)	362 (61.8)	103 (51.5)	
2	169 (21.5)	109 (18.6)	60 (30.0)	
>2	83 (10.6)	60 (10.2)	23 (11.5)	
Drug-resistant epilepsy	210 (27.0)	139 (24.0)	71 (35.9)	0.001 *
History of epilepsy surgery	36 (4.78)	28 (4.8)	8 (4.1)	0.671

SD = standard deviation, ASM = anti-seizure medication; ^a^ Characteristics of vaccinated and unvaccinated groups were made by student’s *t* test (continuous variables) and Pearson’s *χ*^2^ test (categorical variables); ^b^ Other epilepsy etiologies include infectious (19 (2.4%)), %)), metabolic or toxic (5 (0.6%)) and immune (9 (1.1%)); ^c^ Usual seizure frequency refers to average seizure frequency in 2021; * Statistical significance achieved at *p* < 0.05.

**Table 2 vaccines-12-00593-t002:** Self-reported post-vaccination adverse effects.

Self-Reported Adverse Effects	*n* (%)
No adverse effect reported	215 (36.7)
General adverse effects	
Local injection site reaction	247 (42.2)
Fatigue	106 (18.1)
Fever	65 (11.1)
Headache	49 (8.4)
Myalgia	36 (6.1)
Chills	17 (2.9)
Nausea/vomiting	13 (2.2)
Arthralgia	11 (1.9)
Diarrhea	10 (1.7)
Sore throat	3 (0.5)
Lymphadenopathy	0 (0.0)
Others	25 (4.3)
Epilepsy-related adverse effects	
Subjective increase in minor seizure attacks	15 (2.6)
Subjective increase in major seizure attacks	4 (0.7)

**Table 3 vaccines-12-00593-t003:** Seizure frequency in early (7-day) and delayed (30-day) post-vaccination phases.

Subject Group	Number of Seizures 7 Days before and after First Dose (Mean ± Standard Deviation)					Number of Seizures 7 Days before and after Second Dose (Mean ± Standard Deviation)				
	*n*	Pre-Vaccine	Post-Vaccine	*t*	*p*	*n*	Pre-Vaccine	Post-Vaccine	*t*	*p*
All subjects	521	0.33 ± 2.01	0.33 ± 1.93	0.107	0.914	402	0.29 ± 1.71	0.26 ± 1.52	1.118	0.264
BNT162b2 ^a^	307	0.45 ± 2.54	0.43 ± 2.30	0.697	0.486	265	0.34 ± 2.03	0.31 ± 1.78	1.193	0.234
CoronaVac ^a^	212	0.17 ± 0.75	0.19 ± 1.21	−0.242	0.809	134	0.18 ± 0.79	0.17 ± 0.79	0.208	0.836
DRE	121	1.08 ± 3.87	0.90 ± 3.46	2.236	0.027 *	95	0.98 ± 3.31	0.87 ± 2.86	1.181	0.241
**Subject group**	**Number of seizures 30 days after first dose and PVSF** **(Mean ± Standard Deviation)**					**Number of seizures 30 days after second dose and PVSF** **(Mean ± Standard Deviation)**				
	***n* ^b^**	**PVSF ^c^**	**Post-Vaccine**	** *t* **	** *p* **	** *n* **	**PVSF**	**Post-Vaccine**	** *t* **	** *p* **
All subjects	489	1.89 ± 9.86	1.31 ± 8.06	3.436	0.001 *	382	1.60 ± 9.32	1.13 ± 6.40	1.921	0.055
BNT162b2 ^a^	291	2.44 ± 12.44	1.89 ± 10.31	2.373	0.018 *	253	2.04 ± 11.33	1.50 ± 7.74	1.493	0.137
CoronaVac ^a^	198	1.03 ± 3.44	0.46 ± 1.78	2.703	0.007 *	128	0.75 ± 2.13	0.42 ± 1.70	2.739	0.007 *
DRE	118	6.18 ± 18.95	4.35 ± 15.58	3.099	0.002 *	89	5.81 ± 18.66	3.89 ± 12.41	1.936	0.056

PVSF = Pre-vaccination seizure frequency, DRE = Drug-resistant epilepsy; ^a^ No subjects switched between the two vaccines in the first 2 doses. ^b^ Number of subjects with completed PVSF data and with at least one dose of vaccination; ^c^ Pre-vaccination seizure frequency refers to the average monthly seizure frequency in the 3 calendar months prior to the date of first vaccination. * Statistical significance achieved at *p* < 0.05.

**Table 4 vaccines-12-00593-t004:** Univariate analysis of predictors of post-vaccination seizure aggravation (PVSA) ^a^.

Subject Characteristics	No PVSA (*n* = 463) (Mean ± SD, *n* (%))	PVSA (*n* = 26) (Mean ± SD, *n* (%))	OR (95% CI)	*p*
Demographics				
Age, years	51.0 ± 17.9	47.1 ± 14.9	0.99 (0.97–1.01)	0.276
Male sex	257 (55.5)	11 (42.3)	0.59 (0.26–1.31)	0.193
Chinese ethnicity	410 (94.9)	25 (96.2)	1.34 (0.17–10.36)	0.778
Epilepsy characteristics				
Time from diagnosis, years	19.0 ± 13.8	20.4 ± 12.8	1.01 (0.98–1.04)	0.631
Seizure onset				
Focal onset	276 (59.6)	18 (69.2)	1.52 (0.65–3.56)	0.333
Generalized onset	126 (27.2)	7 (26.9)	0.99 (0.40–2.40)	0.974
Unknown/unclassified	77 (16.6)	2 (7.7)	0.42 (0.10–1.80)	0.242
Epilepsy etiology				
Structural	154 (33.3)	9 (34.6)	1.06 (0.46–2.44)	0.887
Genetic	43 (9.3)	1 (3.8)	0.39 (0.05–2.96)	0.363
Others	18 (3.9)	0 (0.0)	-	0.998
Cryptogenic	243 (52.5)	14 (53.8)	1.06 (0.48–2.33)	0.892
Seizure freedom for 3 months	349 (75.4)	6 (23.1)	0.10 (0.04–0.25)	<0.0005 *
Pre-vaccination seizure frequency ^b^				0.054
<1/month	389 (84.2)	17 (65.4)	-	-
≥1/month	35 (7.6)	4 (15.4)	2.62 (0.83–8.20)	0.099
≥1/week	38 (8.2)	5 (19.2)	3.01 (1.05–8.62)	0.040 *
Number of ASMs				0.399
0	25 (5.4)	1 (3.8)	-	
1	306 (66.1)	14 (53.8)	1.14 (0.14–9.06)	0.899
2	87 (18.8)	6 (23.1)	1.72 (0.20–15.00)	0.622
>2	45 (9.7)	5 (19.2)	2.78 (0.31–25.12)	0.363
Drug-resistant epilepsy	106 (23.1)	12 (50.0)	3.32 (1.45–7.61)	0.005 *
History of epilepsy surgery	22 (4.8)	2 (8.3)	1.80 (0.40–8.13)	0.446

PVSA = post-vaccination seizure aggravation, SD = standard deviation, OR = odds ratio, 95% CI = 95% confidence interval, ASM = anti-seizure medication; ^a^ Post-vaccination seizure aggravation is defined as any increase in seizure frequency in the 30 days after each vaccination compared to the baseline period of 3 months before first vaccination; ^b^ Pre-vaccination seizure frequency referred to data in the 3 months before first vaccination only. * Statistical significance achieved at *p* < 0.05.

## Data Availability

The principal authors (W.C.Y.L. and R.W-H.H.) had full access to all the data in the study and take responsibility for its integrity and the data analysis. Anonymized data pertaining to the research presented can be made available upon reasonable request.

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
