# Peer review of "Risk of Seizure Aggravation after COVID-19 Vaccinations in Patients with Epilepsy"

_vaccines, 2024, doi:10.3390/vaccines12060593_

Round 1

Reviewer 1 Report

Comments and Suggestions for Authors

This study is of utmost importance, especially in the context of vaccine hesitancy, which is and still remains a significant public health challenge.

The study found no significant increase in seizure frequency in the early (7 days) or delayed phase (30 days) after vaccination. In addition, seizure frequency was significantly reduced 30 days after vaccination. This finding is crucial as it counters the common concern that vaccinations might increase seizure frequency. Unfortunately, a very strong opinion that patients with epilepsy should not be vaccinated is present even among some physicians.

5.3% of the cohort experienced post-vaccination seizure aggravation, which is in line with previous studies. The presented study emphasizes that those with higher pre-vaccination seizure frequency were at greater risk for PVSA. In contrast, seizure-free patients for three months before vaccination were associated with a lower risk. This is very important information that can guide healthcare providers in managing epilepsy patients and eventually reducing the risk of PVSA through better medication-induced seizure control before vaccination.

The study is also important because it addresses vaccine hesitancy, one of the main problems we vaccinologists never expect to face in the 21st century. Authors also identify key reasons for vaccine hesitancy among epilepsy patients, such as fear of seizure aggravation and perceived lack of scientific evidence—which is all 100% legit. It also points out that healthcare providers discouraged many patients from vaccination, which may have been due to a lack of evidence when the study was conducted.

The study's large cohort size and inclusion of drug-resistant epilepsy patients add to its robustness.

In conclusion, this study provides valuable evidence that can help alleviate concerns about COVID-19 vaccinations among epilepsy patients and support public health efforts to increase vaccination rates.

I would endorse it for publication as it is.

Author Response

Thank you very much for taking the time to review this manuscript. The authors also would like to thank the Reviewer for the positive comments, and for acknowledging the value of the findings in our paper.

Reviewer 2 Report

Comments and Suggestions for Authors

In this manuscript, the author conducted a comprehensive survey using a structured questionnaire to assess the frequency of seizures and vaccination-related adverse effects before and after the administration of COVID-19 vaccine in epilepsy patients. The study demonstrates that there was no significant increase in seizure frequency during the early (7 days) and delayed (30 days) stages post-vaccination, which may provide safety assurance for future COVID-19 vaccinations in epilepsy patients. Before considering publication, the following significant improvements are necessary:

1.  For Table 1.Subject demographics and epilepsy characteristics (n = 786), additional common demographic information such as race, marital status, and educational background should be included. Subsequently, these demographics should be statistically analyzed in relation to COVID-19 vaccination status.

2.  The study extensively utilized univariate logistic regression models. It would be beneficial to include some multivariate analyses to enrich the data and provide stronger evidence that COVID-19 vaccination does not increase the seizure frequency in epilepsy patients.

3.  The authors emphasized that there was no significant increase in seizure frequency post-vaccination. On the contrary, the number of seizures significantly decreased, but they only assessed the effects within a 30-day period. If feasible, it is recommended to continue data collection and extend the longitudinal study duration.

4.  The manuscript included numerous tables. To help readers focus on the critical data, some of the significant positive findings could be integrated into a single figure.

Comments on the Quality of English Language

The author needs to integrate some sentences to make it easier for the reader to read.

Author Response

Thank you very much for taking the time to review this manuscript. Please find the detailed responses below and the corresponding revisions highlighted in the re-submitted files.

Comment 1:
For Table 1. Subject demographics and epilepsy characteristics (n = 786), additional common demographic information such as race, marital status, and educational background should be included. Subsequently, these demographics should be statistically analyzed in relation to COVID-19 vaccination status.

Response 1:
Thank you for the suggestion. Ethnicity data has been added to Table 1 (demographics) and Table 4 (univariate analysis). The manuscript has also been amended accordingly: “The majority of the cohort was of Chinese ethnicity (96.0%). (Lines 156-157)” and “The vaccinated and unvaccinated groups did not differ significantly in terms of age, sex, and ethnicity. (Lines 176-177)”.

Although we agree that it would be interesting to study the correlation between educational background and marital status with vaccination status, this data is not available in our cohort. We have included this as a limitation of this study. “Third, some social demographic data, including educational background and marital status, were not available in our data and may be explored in future studies. (Lines 335-337)”.

Comment 2:
The study extensively utilized univariate logistic regression models. It would be beneficial to include some multivariate analyses to enrich the data and provide stronger evidence that COVID-19 vaccination does not increase the seizure frequency in epilepsy patients.

Response 2:
Thank you very much for the suggestion. We performed multivariate analysis by entering all variables with a univariable p<0.1 using a backward stepwise elimination approach (Lines 144-146). This showed that seizure freedom for 3 months was independently associated with a lower risk of PVSA (OR 0.11, 95% CI 0.04 – 0.28; p<0.0005). (Lines 256-258)

We have added this finding in the abstract (Lines 32-34), discussion (Lines 294-295), and conclusion (Lines 349-351).

Comment 3:
The authors emphasized that there was no significant increase in seizure frequency post-vaccination. On the contrary, the number of seizures significantly decreased, but they only assessed the effects within a 30-day period. If feasible, it is recommended to continue data collection and extend the longitudinal study duration.

Response 3:
Thank you for the suggestion. The scope of our study focused on early (7 days) and delayed (30 days) changes in seizure frequency after vaccination. We agree that further extended longitudinal studies may help determine any long-term change in seizure control among patients with epilepsy. We have included this as a limitation (Line 341-343), and as a suggestion for future studies. 

Comment 4:
The manuscript included numerous tables. To help readers focus on the critical data, some of the significant positive findings could be integrated into a single figure.

Response 5:
Thank you for the suggestion. We have merged Tables 1&2 and Tables 3&4 in the manuscript. 

Comments on the Quality of English Language:
The author needs to integrate some sentences to make it easier for the reader to read.

Response:
Thank you for the comments. We have simplified some sentences to improve their clarity for the readers (Lines 265-267, 306-310).